# Training-free Guidance in Text-to-Video Generation via Multimodal Planning and Structured Noise Initialization

## Abstract

Recent advancements in text-to-video (T2V) diffusion models have significantly enhanced the visual quality of the generated videos. However, even recent T2V models find it challenging to follow text descriptions accurately, especially when the prompt requires accurate control of spatial layouts or object trajectories. A recent line of research uses layout guidance for T2V models that require fine-tuning or iterative manipulation of the attention map during inference time. This significantly increases the memory requirement, making it difficult to adopt a large T2V model as a backbone. To address this, we introduce **VIDEO-MSG**, a training-free Guidance method for T2V generation based on Multimodal planning and Structured noise initialization. VIDEO-MSG consists of three steps, where in the first two steps, VIDEO-MSG creates VIDEO SKETCH, a fine-grained spatio-temporal plan for the final video, specifying background, foreground, and object trajectories, in the form of draft video frames. In the last step, VIDEO-MSG guides a downstream T2V diffusion model with VIDEO SKETCH through noise inversion and denoising. Notably, VIDEO-MSG does not need fine-tuning or attention manipulation with additional memory during inference time, making it easier to adopt large T2V models. VIDEO-MSG demonstrates its effectiveness in enhancing text alignment with multiple T2V backbones (VideoCrafter2 and CogVideoX-5B) on popular T2V generation benchmarks (T2VCompBench and VBench). We provide comprehensive ablation studies about noise inversion ratio, different background generators, background object detection, and foreground object segmentation.[1]

## 1 Introduction

Recent advances in text-to-video (T2V) diffusion models (Yang et al., 2024b; Team, 2025; Kong et al., 2024; Runway, 2024; Agarwal et al., 2025; Bruce et al., 2024; Team, 2024b; Parker-Holder et al., 2024; Inc., 2025; Polyak et al., 2025; OpenAI, 2024) have dramatically improved the quality of generated videos in diverse domains. However, even recent T2V generation models still often struggle to follow text descriptions accurately, especially when the prompt requires accurate control of spatial layouts or object trajectories. As illustrated in fig. 1 (b), recent work has studied improving text alignment by providing detailed layout guidance as an additional input to T2V models, such as bounding boxes (Lin et al., 2024; Long et al., 2024; Lian et al., 2023), optical flow (Liao et al., 2025), and object trajectories (Zhang et al., 2024b; Wu et al., 2024a), which are often created from a large language model (LLM). However, since the original T2V models do not understand the layout guidance, these approaches fine-tune the T2V models with layout annotations (Lin et al., 2024; Long et al., 2024) or iteratively manipulating the attention map of T2V models during inference time (Lian et al., 2023). While effective, these techniques substantially increase memory consumption at inference time or require retraining for different T2V backbones, limiting their scalability to large T2V models.

To address this, we introduce **VIDEO-MSG**, **M**ultimodal **S**ketch **G**uidance for video generation, a training-free guidance method for T2V generation based on multimodal planning and structured

---

[1]We provide code in the supplementary materials.

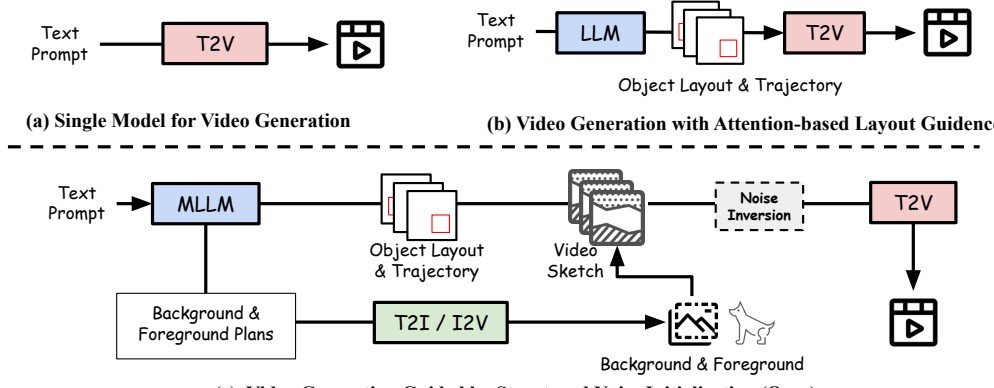

Figure 1: Comparison of different text-to-video generation methods: **(a) single model for video generation**, **(b) video generation with (attention-based) layout guidance**, and our **(c) VIDEO-MSG**, a training-free guidance method for T2V generation based on multimodal planning and structured noise initialization. Since VIDEO-MSG does not need fine-tuning or additional memory during inference time, it is easier to adopt large T2V models than previous video layout guidance methods based on fine-tuning or iterative attention manipulation.

noise initialization, as illustrated in fig. 1 (c). As illustrated in fig. 2, VIDEO-MSG consists of three steps: (1) background planning (section 3.1), (2) foreground object layout and trajectory planning (section 3.2), and (3) video generation with structured noise inversion (section 3.3). From the first two steps, VIDEO-MSG creates VIDEO SKETCH, a fine-grained spatial and temporal plan with a set of multimodal models, including multimodal LLM (MLLM), object detection, and instance segmentation models. Then in the last step, VIDEO-MSG guides a downstream T2V diffusion model with VIDEO SKETCH through structured noise inversion and denoising. Notably, VIDEO-MSG does not need fine-tuning or additional memory during inference time, making it easier to adopt large T2V models, compared to existing methods based on fine-tuning or iterative attention manipulation.

VIDEO-MSG demonstrates their effectiveness in enhancing text alignment with multiple T2V backbones (VideoCrafter2 (Chen et al., 2024) and CogVideoX-5B (Yang et al., 2024b)) on popular T2V generation benchmarks (T2VCompBench (Sun et al., 2024) and VBench (Huang et al., 2024)). For example, VIDEO-MSG improves motion binding with a relative gain of 52.46%, numeracy with a relative gain of 40.11%, and spatial relationship with a relative gain of 11.15% with CogVideoX-5B as a T2V generation backbone. We provide comprehensive quantitative and qualitative ablation studies about noise inversion ratio, different background generators, background object detection, and foreground object segmentation. We hope our method can inspire future work on effectively and efficiently integrating LLMs' planning ability into video generation.

## 2 RELATED WORK

### 2.1 MLLM PLANNING FOR VIDEO GENERATION

There are recent research works (Wang et al., 2024c; Lin et al., 2024; Lian et al., 2024; He et al., 2023; Zhou et al., 2024b) that leverage the reasoning capabilities and world knowledge of LLMs or multimodal LLMs for the task of video generation. For example, one line of work (Lin et al., 2024; Wang et al., 2024c; Lian et al., 2024) applies GPT-4 / GPT-4o to expand a single text prompt into a 'video plan' in the format of bounding boxes or detailed prompt description (Yang et al., 2024a), which is then given as input to downstream video diffusion model for layout-guided video generation. The other line of work (Kondratyuk et al., 2024; Tong et al., 2024; Wang et al., 2024a; Wu et al., 2024b; Lu et al., 2023; 2024) performs token-level planning utilizing multimodal LLMs. For example, (Wang et al., 2024a; Kondratyuk et al., 2024) tokenize videos and text into the same space and generate video tokens using the same strategy as text (e.g., next-token prediction). However, both directions either rely on high-quality prompts and the bounding box planning or require extensive training and do not fully leverage the power of existing visual tools for fine-grained video generation.

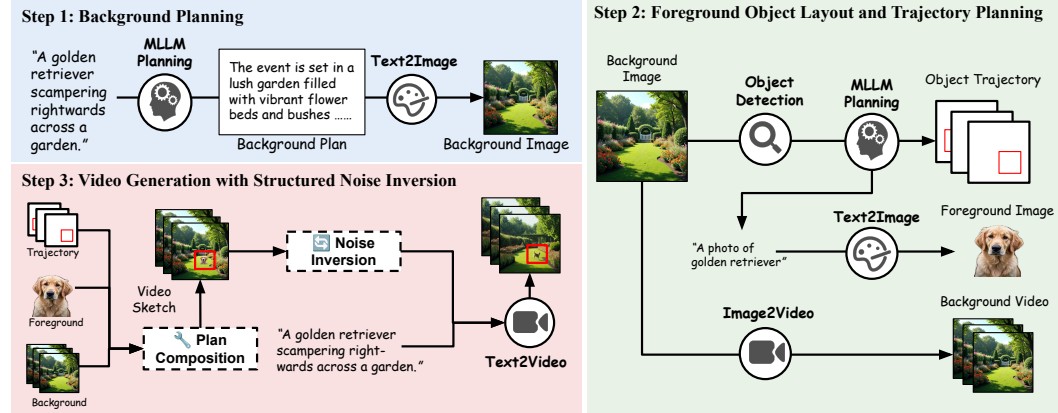

Figure 2: Three stages of VIDEO-MSG. In the first stage, the MLLM plans specific global and local contexts that fit the provided text-to-video prompt. The text-to-image (T2I) model uses the MLLM planned context to render the necessary components of the video. In the third stage, we generate video with VIDEO SKETCH via noise inversion.

In contrast, our work leverages the power of both multimodal LLMs and image/video diffusion models to generate a VIDEO SKETCH for final fine-grained motion control, and is fully training-free.

## 2.2 MOTION DIRECTION CONTROL IN VIDEO GENERATION

Controllability in video generation is gaining increasing attention in the field of generative AI, as it enables models to generate videos aligned with user intent. One line of research focuses on training models with the capability of trajectory control, camera control, or motion control by generating intermediate representations. For trajectory control, recent works such as DragNUMA (Yin et al., 2023), IVA-0 (Yu et al., 2024), DragAnything (Wu et al., 2024a), and TrackGo (Zhou et al., 2024a) encode object movement trajectories into dense features, which are then fused into the diffusion model to enable object movement control. On the other hand, CameraCtrl (He et al., 2024), MotionCtrl (Wang et al., 2024b), and Image Conductor (Li et al., 2024) encode camera extrinsics as features to control camera motion in the generated videos. A common drawback of both of these categories is their reliance on accurate object trajectory or camera movement information, which are difficult for users to manipulate directly. Additionally, video datasets with accurate trajectory annotations are limited, which constrains the performance of these models. The third category, including VideoJAM (Chefer et al., 2025) and MotionI2V (Shi et al., 2024), produces motion and video representations jointly, or sequentially by first generating intermediate representations, which then serve as guidance for generating video outputs. However, such methods require extensive training due to extra generation objectives. In contrast, our method uses an image-to-video model, allowing us to transform existing, real-world images into controllable videos under LLM planning.

## 3 METHOD

We introduce **VIDEO-MSG**, **M**ultimodal **S**ketch **G**uidance for video generation, a training-free guidance method for T2V generation based on multimodal planning and structured noise initialization. VIDEO-MSG consists of three stages (illustrated in fig. 2):

- **Background planning** (section 3.1), where we adopt T2I and I2V models to generate background image priors with natural animation.
- **Foreground Object Layout and Trajectory Planning** (section 3.2), where we apply MLLM and object detectors to plan and place foreground objects into the background harmoniously.
- **Video Generation with Structured Noise Initialization** (section 3.3), where the synthesized images derived from the above stages are used as VIDEO SKETCH for final video generation via inversion techniques.

## 3.1 Background Planning

Given a prompt for video generation, we first ask an MLLM (GPT-4o (OpenAI et al., 2024)) to generate a detailed background description (see Stage 1 in fig. 2). Here, we explicitly instruct the MLLM to generate only the background and avoid including any moving or key objects mentioned in the original prompt, thereby enforcing proper decoupling. We find that this strategy helps address issues in conditional T2I generation based on bounding boxes, where the T2I model may fail to generate the foreground object at the specified box location in the image. In addition, we explore two approaches for background generation:

(1) Using a T2I model to generate an initial background, followed by an I2V model to animate it. In this way, we can adopt a strong T2V model to potentially achieve improved video aesthetic quality.
(2) Directly using a T2V model to generate the background with animation, which avoids the potential distribution gap between the two models in (1).

In both cases, we adopt a video generation model. We aim to introduce natural background animation rather than keeping it static while only animating foreground objects. This ensures that elements such as flowing water, moving clouds, or swaying trees are naturally incorporated, making the generated videos more realistic and visually coherent. Moreover, by comparing approaches (1) with (2), we notice that the advantage of adopting a strong T2I model in (1) outweighs the domain gap between the T2I and I2V models in (2) as discussed in Sec. 4.3. Therefore, we apply approach (1) as our default experiment setting.

## 3.2 Foreground Object Layout and Trajectory Planning

This stage aims to place the property of the foreground object in the background in a spatially coherent manner. We first implement this stage by providing the background images generated in stage 1, along with a prompt describing movement dynamics to GPT-4o (OpenAI et al., 2024), then ask it to generate a sequence of bounding boxes to represent the foreground object's movement. For instance, given the text prompt: "A cat sinking to the left in the living room", GPT-4o can correctly infer the cat's movement direction (i.e., moving left). However, when provided with a background image of a living room, GPT-4o often fails to position the cat's bounding box appropriately on the floor (e.g., with the bounding box floating in mid-air or overlapping with unrelated objects), as illustrated in Figure 4a. This suggests that while GPT-4o demonstrates strong motion reasoning capabilities, it lacks direct grounding capability for visual elements and struggles to align foreground objects with the background scene in a spatially consistent manner.

To overcome this limitation, we first detect all objects in the background image with Recognize-Anything (RAM) (Zhang et al., 2024a) then extract their bounding boxes with Grounding-DINO (Liu et al., 2023). These bounding boxes are fed into GPT-4o to provide explicit spatial context, which helps it accurately position and animate foreground objects, enhancing spatial coherence in generated videos and reducing placement errors. Qualitative examples of the effectiveness of object detection with Grounding-DINO and RAM are presented in Figure 4a. With the above inputs (i.e.,, video text prompt, background image, and the bounding boxes of objects in the background), GPT-4o generates a sequence of bounding boxes for the foreground objects in the format [object name, bounding box coordinates] (see stage 2 in fig. 2). Additionally, it provides a textual description for each frame and a reasoning process explaining the planned object motions after the sequence of frames. This reasoning step enhances the coherence and accuracy of motion planning.

Once the sequence of bounding boxes is obtained, we utilize a T2I model to generate the appearance of the foreground object using the prompt: "An image of {object name}." However, directly merging the generated object image with the background presents a challenge—the background in the generated object image can significantly affect the overall visual coherence, as illustrated in fig. 4b. To address this, we apply SAM (Kirillov et al., 2023) to extract the object from the generated image, removing any unintended background. Based on the planned bounding boxes, the extracted object is then resized and placed onto the background image at the corresponding location. This process ensures a more seamless integration of the foreground object into the background, improving the visual consistency of the generated video.

## 3.3 Video Generation with Structured Noise Initialization

In this stage, we generate a final video by guiding the T2V diffusion model with the VIDEO SKETCH created from the previous stage (section 3.2). Inversion methods (Song et al., 2020), which are often used in image and video editing tasks (Sheynin et al., 2023; Meng et al., 2022), can be effectively utilized here to create structured noise to fuse the information from VIDEO SKETCH. While the normal denoising process starts from the terminal timestep $t = T$ (a random noise) to the initial timestep $t = 0$ (a clean video), we create per-frame initial noises from VIDEO SKETCH via noise inversion (Meng et al., 2022) and start denoising from a timestep $t^{\text{inv}}$. Specifically, we first encode the sequence of VIDEO SKETCH frames into the latent space $z$ using a 3D VAE (Yang et al., 2024b; Kingma & Welling, 2014). Next, we obtain the initial noise $z_{t^{\text{inv}}}$ via the forward diffusion process (Ho et al., 2020): $z_{t^{\text{inv}}} = \sqrt{\alpha_t} z_0 + \sqrt{1 - \alpha_{t^{\text{inv}}}} \epsilon, \quad \epsilon \sim \mathcal{N}(0, I)$, where $\alpha_t = \prod_{s=1}^{t}(1 - \beta_s)$ is the cumulative noise schedule, and $\epsilon$ represents Gaussian noise. We parameterize $t^{\text{inv}} = \alpha \times T$, where $\alpha \in (0.0, 1.0)$. Inspired by VideoDirectorGPT (Lin et al., 2024), which uses an LLM to estimate a confidence score along with bounding box layouts as layout guidance strength, we employ an LLM to infer an appropriate noise inversion ratio $\alpha$ value given a text description. (see section 4.3 for detailed experiments). We explain more details about the noise inversion in Appendix.

# 4 Experiments

## 4.1 Experiment Setups

**Datasets.** We evaluate VIDEO-MSG on popular text-to-video generation benchmarks, T2V-CompBench (Sun et al., 2024) and VBench (Huang et al., 2024). T2V-CompBench and VBench measure diverse aspects of text-to-video generation tasks with seven (e.g., consistent attribute binding, motion binding, spatial relationships) and sixteen categories (e.g., overall consistency, color, temporal flickering, motion smoothness), respectively. In this work, we primarily use T2V-CompBench to evaluate video diffusion models' capability in compositional text-to-video generation, and use VBench to measure the motion smoothness of the generated video.

**Implementation details.** We implement VIDEO-MSG on two recent text-to-video generation diffusion models: VideoCrafter2 (Chen et al., 2024) and CogVideoX-5B (Yang et al., 2024b). To generate the VIDEO SKETCH, we employ FLUX.1-dev (Labs, 2024) and SDXL (Podell et al., 2024) as the background generator, and CogVideoX-5B as the image-to-video generator. We utilize Recognize-Anything (Zhang et al., 2024a) and Grounded-Segment-Anything (Kirillov et al., 2023) for foreground object segmentation. We utilize GPT4o as the multi-modal LLM for background description generation, foreground object layout and trajectory planning, and determining the noise inversion ratio $\alpha$ dynamically based on the prompt. For noise inversion ratio $\alpha$ (section 3.3), we find the range [0.7, 0.9] works well for CogVideoX-5B, and the range [0.5, 0.8] works well for VideoCrafter2 (see section 4.3 for ablation study). All experiments are conducted on A100 and A6000 GPUs, with batch size 1 and an approximate memory usage of 16 GB. We provide additional details, such as prompts used for GPT-4o, in the Appendix.

## 4.2 Quantitative Evaluation

**Improved control on spatial layout and object trajectory.** Table 1 shows that VIDEO-MSG significantly improves both T2V backbone models (VideoCrafter2 and CogVideoX-5B) in many skills, especially in motion binding ('Motion'), with an increase of 0.1499 on VideoCrafter2 and 0.1544 on CogVideoX-5B. VIDEO-MSG also provides large improvements in spatial relationships ('Spatial'), and numeracy ('Numeracy') in both backbone models. These results show that the planning and structured noise initialization of VIDEO-MSG effectively improve the control of spatial layouts and object trajectories in video generation. It is also noteworthy that VIDEO-MSG, implemented with open-source T2V backbone models, archives higher motion binding scores than closed-source models such as Gen-3 (Runway, 2024). The VIDEO-MSG did not improve the scores in dynamic attribute binding ('Dynamic-attr') and object action and interaction ('Action' and 'Interaction') categories. This is likely because dynamic changes in object or environment states and interactions and actions between objects are difficult to guide solely with bounding boxes.

| Model | Consist-attr | Dynamic-attr | Spatial | Motion | Action | Interaction | Numeracy |
|---|---|---|---|---|---|---|---|
| *(Closed-source models)* | | | | | | | |
| Pika (Team, 2024a) | 0.6513 | 0.1744 | 0.5043 | 0.2221 | 0.5380 | 0.6625 | 0.2613 |
| Gen-3 (Runway, 2024) | 0.7045 | 0.2078 | 0.5533 | 0.3111 | 0.6280 | 0.7900 | 0.2169 |
| Dreamina (Dreamina AI, 2024) | 0.8220 | 0.2114 | 0.6083 | 0.2391 | 0.6660 | 0.8175 | 0.4006 |
| PixVerse (PixVerse, 2024) | 0.7370 | 0.1738 | 0.5874 | 0.2178 | 0.6960 | 0.8275 | 0.3281 |
| Kling (Kling AI, 2024) | 0.8045 | 0.2256 | 0.6150 | 0.2448 | 0.6460 | 0.8475 | 0.3044 |
| *(Open-source models)* | | | | | | | |
| ModelScope (Wang et al., 2023) | 0.5483 | 0.1654 | 0.4220 | 0.2552 | 0.4880 | 0.7075 | 0.2066 |
| ZeroScope (Sterling, 2024) | 0.4495 | 0.1086 | 0.4073 | 0.2319 | 0.4620 | 0.5550 | 0.2378 |
| AnimateDiff (Guo et al., 2023) | 0.4883 | 0.1764 | 0.3883 | 0.2236 | 0.4140 | 0.6550 | 0.0884 |
| Latte (Ma et al., 2024) | 0.5325 | 0.1598 | 0.4476 | 0.2187 | 0.5200 | 0.6625 | 0.2187 |
| Show-1 (Zhang et al., 2023) | 0.6388 | 0.1828 | 0.4649 | 0.2316 | 0.4940 | 0.7700 | 0.1644 |
| Open-Sora 1.2 (hpcaitech, 2024) | 0.6600 | 0.1714 | 0.5406 | 0.2388 | 0.5717 | 0.7400 | 0.2556 |
| Open-Sora-Plan v1.1.0 (Lab & etc., 2024) | 0.7413 | 0.1770 | 0.5587 | 0.2187 | **0.6780** | 0.7275 | 0.2928 |
| VideoTetris (Tian et al., 2024) | 0.7125 | 0.2066 | 0.5148 | 0.2204 | 0.5280 | 0.7600 | 0.2609 |
| Vico (Yang & Wang, 2024) | 0.7025 | **0.2376** | 0.4952 | 0.2225 | 0.5480 | 0.7775 | 0.2116 |
| VideoCrafter2 (Chen et al., 2024) | 0.6750 | 0.1850 | 0.4891 | 0.2233 | 0.5800 | 0.7600 | 0.2041 |
| VideoCrafter2 + LVD (Lian et al., 2023) | 0.6663 | 0.2308 | 0.5106 | 0.2178 | 0.5640 | 0.8125 | 0.2869 |
| | (-0.0087) | (+0.0458) | (+0.0215) | (-0.0055) | (-0.0160) | (+0.0525) | (+0.0828) |
| VideoCrafter2 + VIDEO-MSG (Ours) | **0.7536** | 0.2110 | 0.5866 | 0.3732 | 0.5737 | **0.8220** | 0.3138 |
| | (+0.0786) | (+0.0260) | (+0.0975) | (+0.1499) | (-0.0063) | (+0.0620) | (+0.1097) |
| CogVideoX-5B (Yang et al., 2024b) | 0.7220 | 0.2334 | 0.5461 | 0.2943 | 0.5960 | 0.7950 | 0.2603 |
| CogVideoX-5B + VIDEO-MSG (Ours) | 0.7109 | 0.2102 | **0.6070** | **0.4487** | 0.5960 | 0.7800 | **0.3647** |
| | (-0.0111) | (-0.0232) | (+0.0609) | (+0.1544) | (+0.0000) | (-0.0150) | (+0.1044) |

Table 1: **T2V-CompBench evaluation results**. We highlight the best/second-best scores for open-sourced models with **bold**/underline.

| No. | Noise inversion ratio $\alpha$ | T2V-CompBench | | | VBench |
|---|---|---|---|---|---|
| | | Motion Binding | Numeracy | Spatial | Motion Smoothness |
| 1. | Direct T2V (no inversion) | 0.2233 | 0.2041 | 0.4891 | 97.73 |
| 2. | 0.8 | 0.2793 | 0.2081 | 0.5502 | 98.69 |
| 3. | 0.7 | 0.3197 | 0.2653 | 0.5678 | 98.62 |
| 4. | 0.6 | 0.3352 | 0.3059 | 0.6057 | 98.63 |
| 5. | 0.5 | **0.3980** | **0.3138** | **0.6447** | 98.58 |
| 6. | LLM-controlled | 0.3732 | **0.3138** | 0.5866 | **99.01** |

Table 2: Comparison of different noise inversion ratio $\alpha$, where we compare static values and LLM-based dynamic values. Backbone T2V: VideoCrafter2. Background generator: Flux + CogVideoX-5B.

**Comparison to planning-based baseline.** We also compare VIDEO-MSG with LVD (Lian et al., 2023), a recent T2V layout guidance method, where it adds a gradient-based energy function optimization step before each denoising step of the T2V diffusion backbone. The energy function adjusts the cross-attention map of diffusion models to concentrate within a set of object bounding boxes generated by an LLM. On the VideoCrafter2 backbone, we find that VIDEO-MSG outperforms LVD in all categories except for dynamic attribute binding, with the largest improvement observed in motion binding ('Motion'), where VIDEO-MSG surpasses LVD by 0.1554. This demonstrates the effectiveness of our approach. Note that VIDEO-MSG is also more memory-efficient than LVD, as the layout guidance in LVD requires backpropagation through the T2V diffusion backbone, making it hard to adapt to large diffusion models; we could implement VIDEO-MSG with CogVideoX-5B backbone to run on an A6000 GPU (48GB), but we could not fit LVD even on an A100 (80GB).

### 4.3 ABLATION STUDIES

**Noise inversion ratio $\alpha$.** As described in section 3.3, we guide the T2V generation backbone by denoising from an intermediate timestep $t^{\text{inv}} = \alpha \times T$ to the initial timestep $t = 0$. Here, we experiment with different noise inversion ratios $\alpha$ (i.e., varying the noise injected into the VIDEO SKETCH). Table 2 shows that lower $\alpha$ achieves better performance in motion binding (e.g., moving left/right), numeracy, and spatial relationships but hurts the smoothness of motions. This aligns with the intuition that increasing the number of refinement steps based on VIDEO SKETCH enhances the final motion quality. We observe that automatically inferring proper $\alpha$ given text description with LLM achieves a good trade-off and use this approach by default.

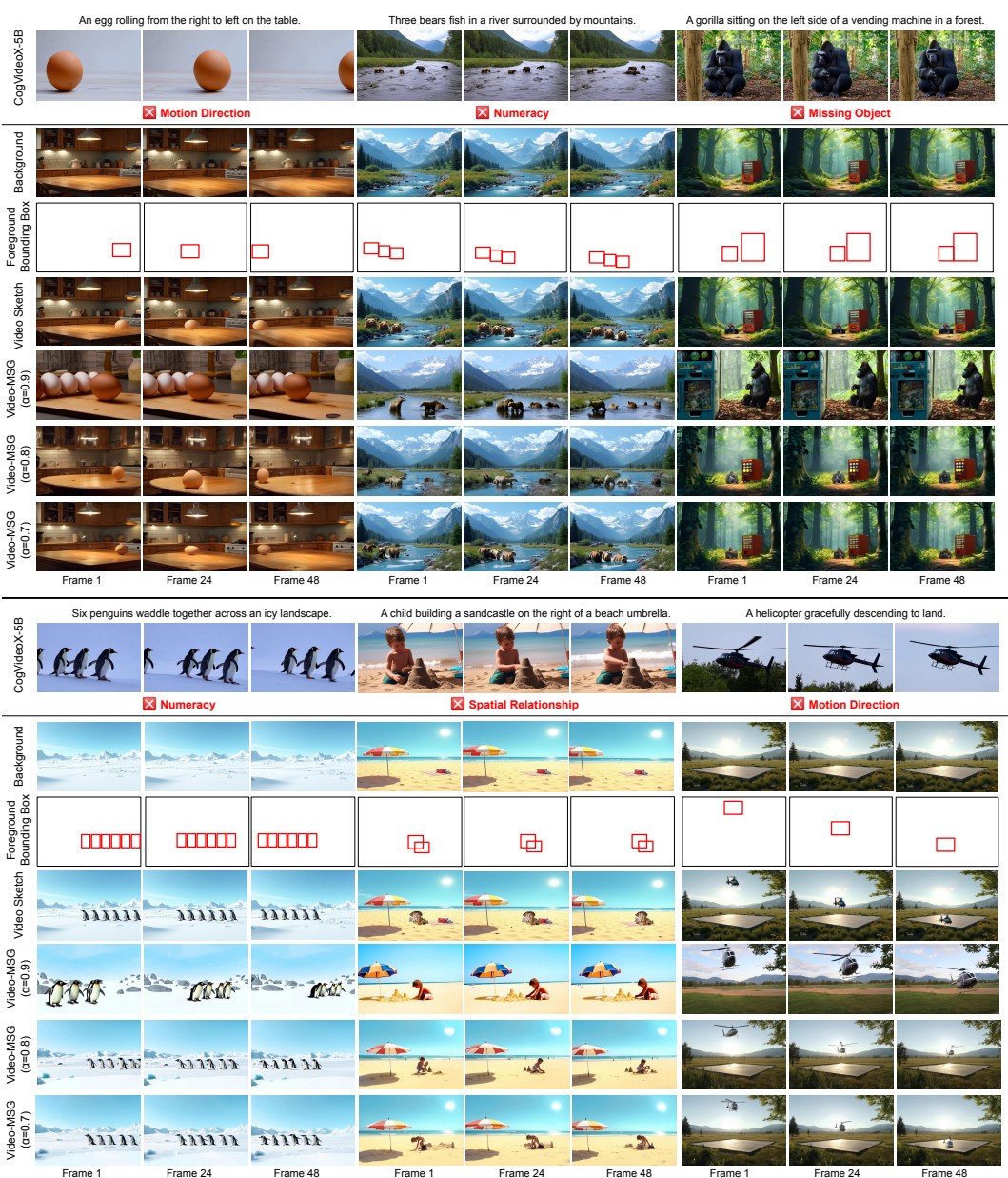

Figure 3: Videos generated with CogVideoX-5B and VIDEO-MSG with CogVideoX-5B backbone. The videos generated with VIDEO-MSG are more accurate regarding object motions, numeracy, and spatial relationships.

**Different background generator.** In Table 3, we compare different background generation methods (section 3.1): (1) generating a background with a text-to-image (T2I) model, followed by an image-to-video (I2V) model for animation, and (2) directly using a text-to-video (T2V) model to generate an animated background. While both approaches improve motion binding and numeracy compared to using a single T2V model for video generation, the T2I + I2V pipeline scores higher in numeracy, and the T2V

| No. | Background Generator | Motion | Numeracy |
|-----|---------------------|--------|----------|
| 1. | Direct T2V (no background) | 0.2897 | 0.2750 |
| 2. | SDXL (T2I) + CogVideoX-5B (I2V) | 0.4487 | 0.3559 |
| 3. | FLUX (T2I) + CogVideoX-5B (I2V) | 0.4549 | **0.3647** |
| 4. | CogVideoX-5B (T2V) | **0.4565** | 0.3028 |

Table 3: Ablation studies on different background generators.

approach scores higher in motion binding. We attribute this to the video generation model's ability to better refine object motion when the background follows a static camera, making foreground changes more salient for the video diffusion model. The I2V pipeline better adheres to the "Static Camera" prompt, producing natural background animations (e.g., wind, light changes). In contrast, T2V models often disregard the "Static Camera" requirement, introducing excessive camera motion and scene changes in the video. These inconsistencies make it harder for the video diffusion model to refine foreground objects, leading to performance degradation (e.g., 0.3028 with CogVideoX-5B vs. 0.3647 with FLUX on numeracy). Additionally, we find that a stronger T2I model (e.g., FLUX (Labs, 2024)) yields better results than a weaker one (e.g., SDXL (Podell et al., 2024)), highlighting the potential of leveraging high-quality T2I models for layout-controlled text-to-video generation.

**Different LLM planner.** In Table 4, we compare different LLMs for generating foreground and background plans: GPT-4o and Qwen2-VL-7B-Instruct (Bai et al., 2023). In both settings, we use VideoCrafter2 as the video generator, and 0.5 as the noise inversion ratio. Our results show that stronger LLMs produce plans that align more closely with the input prompts, and high-quality plans significantly enhance performance. Specifically, for spatial relationship, using plans generated by GPT-4o improves the baseline by 0.1556, whereas Qwen2-VL-7B-Instruct yields only a 0.0275 improvement. These results

| No. | LLM Planner | Spatial |
|-----|-------------|---------|
| 1. | Direct T2V (no LLM) | 0.4891 |
| 2. | Qwen2-VL-7B-Instruct | 0.5166 |
| 3. | GPT4-o | **0.6447** |

Table 4: Ablation studies on different llm planners. Backbone T2V: VideoCrafter2. Noise inversion ratio: 0.5.

highlight the importance of plan quality and suggest that leveraging more powerful LLMs can further boost performance.

## 4.4 QUALITATIVE ANALYSIS

**VIDEO SKETCH improves control of spatial layout and object trajectory.** fig. 3 compares videos generated from CogVideoX-5B, and VIDEO-MSG (with CogVideoX-5B backbone). We observe that CogVideoX-5B struggles with motion direction (e.g., an egg moves to the right instead of to the left, a helicopter ascends instead of descending to the land), numeracy (e.g., generated four bears instead of three bears, four penguins instead of six penguins), and spatial relationships (e.g., a vending machine should be located to the right of a gorilla, but it is missing; the umbrella should be located on the left of the children). In contrast, VIDEO-MSG successfully guides the T2V backbone to generate videos with correct semantics in all cases. Note that the T2V model can understand the coarse guidance in VIDEO SKETCH and place objects that harmonize well with the background through noise inversion. For example, in the middle example ('three bears in a river surrounded by mountains'), even when the VIDEO SKETCH includes three bears only with other heads facing forward, the T2V model could place the three bears in the river naturally.

**Effect of different noise inversion ratios $\alpha$.** fig. 3 shows the video generation results from VIDEO SKETCH (with CogVideoX-5B backbone), with different noise inversion ratios $\alpha$. Interestingly, the model can automatically refine objects to better align with the prompt and surrounding environment based on different $\alpha$. We find that lower $\alpha$ (i.e., less noise) generally provides stronger layout control. For example, in the left top example, the egg in the videos with $\alpha = 0.7$ and $\alpha = 0.8$ closely follow the trajectory in VIDEO SKETCH, while in the video $\alpha = 0.9$, the egg movement is small and does not follow the trajectory. However, a lower $\alpha$ can lead to less natural generations; e.g., in the bottom-middle example, the boy motion appears less natural at $\alpha = 0.7$ compared to $\alpha = 0.9$. This highlights the importance of selecting an appropriate $\alpha$ to balance motion smoothness with faithful adherence to VIDEO SKETCH.

**Background object detection helps foreground object placement.** We find that a deep understanding of the background images through object detection is crucial in foreground planning (section 3.2). As shown in fig. 4a, without access to background bounding box information, the MLLM fails to place the golden retriever on the grass when relying solely on the background image input. In contrast, when provided with bounding information from the background (e.g., {"label": "path", "box": [0.44, 0.57, 0.99, 0.99]}), the MLLM successfully positions the golden re-

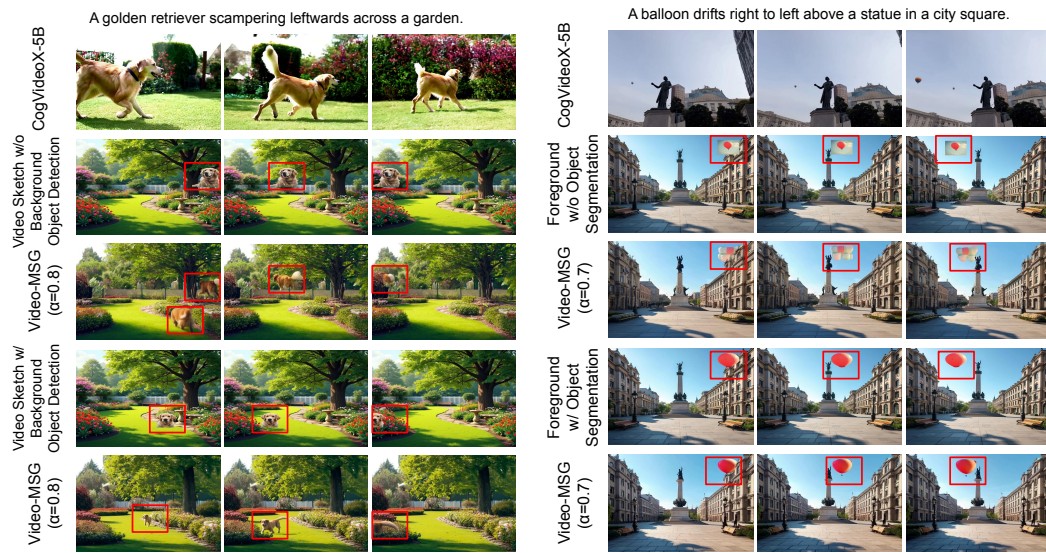

(a) Example video showing the importance of background object detection in foreground object placement.

(b) Example video showing the importance of foreground object segmentation.

Figure 4: Qualitative examples demonstrating the importance of background object detection and foreground object segmentation.

triever at the correct location on the grass. Moreover, we find that this planning step directly impacts the final video quality. Conditioning the generation on inaccurate bounding box plans can conflict with the video diffusion model's prior knowledge. For instance, in the first frame, the model may generate two golden retrievers—one on the ground based on its prior knowledge and another floating in the air according to the VIDEO SKETCH —resulting in unrealistic outputs, such as a golden retriever running mid-air across the garden. In contrast, our approach, which conditions planning on background bounding boxes, enables the generation of more natural and commonsense-aligned videos.

**Segmentation of foreground objects improves harmonization.** As demonstrated in fig. 4b, without object segmentation, the foreground object (a balloon) does not align well with the background, and the quantity is not well controlled (multiple balloons). This occurs because the video diffusion model does not inherently distinguish between the appearance of the background and that of the balloon, causing them to blend together. In contrast, when we first segment the balloon from the generated foreground object image and then place it onto the background to create VIDEO SKETCH, the balloon in the generated video harmonizes well with the background.

## 5 CONCLUSION

In this work, we introduce VIDEO-MSG, a training-free guidance method designed to enhance text-to-video (T2V) generation through multimodal planning and structured noise initialization. VIDEO-MSG consists of three steps, wherein the first two steps, VIDEO-MSG creates VIDEO SKETCH, a detailed spatial and temporal plan utilizing a set of multimodal models, including multimodal LLM, object detection, and instance segmentation models. In the final step, VIDEO-MSG guides a downstream T2V diffusion model with VIDEO SKETCH through noise inversion and denoising. Notably, VIDEO-MSG does not require fine-tuning or additional memory during inference, making it easier to adopt large T2V models than existing methods that rely on fine-tuning or iterative attention manipulation. VIDEO-MSG demonstrates its effectiveness in enhancing text alignment with multiple T2V backbones on popular T2V generation benchmarks. We also provide comprehensive ablation studies and qualitative examples that support the design choices of VIDEO-MSG. We hope our method can inspire future work on effectively and efficiently integrating LLMs' planning capabilities into video generation.

ETHICS STATEMENTS

We do not foresee any ethical implications beyond standard ethical and safety considerations that apply to AI research generally.

REPRODUCIBILITY STATEMENT

We provide the code in the supplementary materials. All the implementation details can be found in Sec 4. All prompts used can be found in Appendix B.

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

## APPENDIX

In this appendix, we present the following:

- Noise inversion details using DPM-Solver++ (Lu et al., 2022) in Sec. A.
- MLLM prompts we use to collect background description, foreground object layout and trajectory, and $\alpha$ used to determine how much noise to inject during inversion in Sec. B.
- Limitations, broader impact and licenses are discussed in Sec. C and Sec. D.

> You need to provide a detailed description of the background for a given event. The description should be about the environment, lighting, and setting, without including any moving objects or entities. The background description should not include any key objects that are mentioned in the prompt (e.g., don't include objA/objB if the prompt is like objA is left to objB, a white objA beside a green objB). The background description should not exceed 50 words.
>
> Example: {In-Context Learning Example}
>
> User: Provide a background description for the event: {Prompt}

Figure 5: Prompt template used to query background description.

## A  NOISE INVERSION DETAILS

Here, we describe in detail how we adopt the inversion technique for final video generation with the motion priors prepared in Stage 3. Specifically, we first encode the sequence of images with the planned layout, collected in the previous stage, into the latent space $z$ using a 3D Variational Autoencoder (3D VAE). Then, we perform the forward diffusion process where Gaussian noise is gradually added to the latent. Following the DPM-Solver++ (Lu et al., 2022) scheduler in CogVideoX, the noised latent at diffusion step $t$ is:

$$z_t = \sqrt{\alpha_t}z_0 + \sqrt{1 - \alpha_t}\epsilon, \quad \epsilon \sim \mathcal{N}(0, I). \tag{1}$$

Here, $\alpha_t = \prod_{s=1}^{t}(1 - \beta_s)$ is the cumulative noise schedule, and $\epsilon$ represents Gaussian noise.

Then, given a noisy latent $z_t$, we attempt to recover the clean latent as a general reverse denoising starting from step $t$. The model then denoises to $z_{t-1}$ using the DPM-Solver++ method, which provides a high-order approximation of the reverse diffusion process. Specifically, the update equation for $z_{t-1}$ in a second-order solver is:

$$z_{t-1} = z_t + \lambda_1\hat{F}(z_t, t) + \lambda_2\hat{F}(z_t + \lambda_3\hat{F}(z_t, t), t_m), \tag{2}$$

where $\hat{F}(z, t) = -\frac{1}{2}\beta_t z - g^2(t)\epsilon_\theta(z, t)$ is the estimated drift term, $\lambda_1, \lambda_2, \lambda_3$ are step-size coefficients computed adaptively, and $t_m$ is an intermediate timestep between $t$ and $t - 1$.

We observe that selecting $t$ within a specific range enables video diffusion models to inject smooth object motions naturally. This process effectively transforms a sequence of static images into a coherent video with realistic motion dynamics.

## B  PROMPT FOR MLLM PLANNING

In this section, we present the prompt templates used to collect background descriptions, foreground object layouts and trajectories, and the parameter $\alpha$, which determines the amount of noise injected during inversion. As shown in fig. 5, we explicitly instruct the multi-modal LLM to separate the generation of foreground and background, ensuring that key foreground objects are not mistakenly included in the background. In fig. 6, we first prompt the MLLM to generate bounding boxes for foreground objects, followed by reasoning for their placement. Additionally, we emphasize that the placement of foreground objects should be informed by background bounding box annotations to improve spatial coherence. Finally, fig. 7 illustrates the prompt template used to determine the appropriate level of noise injection during inversion. We explicitly incorporate prior knowledge into the template, instructing the MLLM to apply less noise for tasks requiring precise trajectory or layout control and more noise for tasks involving dynamic changes or object actions that cannot be effectively modeled with bounding box plans.

## C  LIMITATIONS AND BROADER IMPACT

Text-to-video generation models have broad real-world applications, such as filmmaking or enhancing existing videos through recapturing. Our proposed method, VIDEO-MSG, enables improved control

Assuming the frame size is normalized to the range 0-1, you need to give a possible 25-frame layout with bounding boxes of the entities of a given event. The background image is provided. Besides, the objects in the background image are also annotated with bounding box, and these objects are static objects across the 25 frames. You need to generate the moving object at the correct location based on the input background image and bounding box annotation. You don't need to include the box plan of the objects that already in the background.
You need to generate plan for all the key objects in the prompt (e.g., Generate both objA and objB for prompt objA is left to objB). There should be enough distance between the two objects so that each object can be generated clearly.
The object name should contain its attributes (e.g., color, shape).
Number the objects if there the multiple same objects in the prompt (e.g., 2 birds).
Each object in the image is one rectangle or square box in the layout and size of boxes should be as large as possible. You need to generate layouts from the close up camera view of the event.
The layout difference between two adjacent frames must be small, considering the small interval.
You need to generate a caption that best describes the image for each frame. After generating all frames, add reasoning to your design.
Please strictly follow the user format here to generate the plan. Do not include any code or calculation before generating the plan. Add all the reasoning process after generating the plan.
Use format:
Frame_1: [[object1, [left, top, right, bottom]], [object2, [left, top, right, bottom]], ..., [object_n, [left, top, right, bottom]]], caption:...
Frame_2: [[object1, [left, top, right, bottom]], [object2, [left, top, right, bottom]], ..., [object_n, [left, top, right, bottom]]], caption:...
...
Frame_25: [[object1, [left, top, right, bottom]], [object2, [left, top, right, bottom]], ..., [object_n, [left, top, right, bottom]]], caption:...
Reasoning:...'

Here's one example:
{In-Context Learning Example}

User Input:
User: Provide bounding box coordinates for the prompt: {Prompt}
Background box annotation: {Background Bounding Box Annotation}
Background image: {Image}

Figure 6: Prompt template used to query foreground object layout and trajectory plan.

You are an assistant tasked with guiding video generation. The video is created based on conditions set by a pre-planned sequence of images. Your role is to determine how much influence these pre-planned images should have on the final video, balanced against the video generation system's own capabilities.
Decision Rules:
1. High control by pre-planned images:
(1) When the prompt including the description of the movement direction of the objects.
(2) When the prompt emphasizes generating multiple objects in the video.
2. Low control by pre-planned images:
(1) When the prompt describes specific actions of objects (e.g., singing), or interaction between the objects, or focuses on the state changes of the background.
(2) When the prompt specifies attributes of objects (e.g., color, shape).
Control Levels: You will select one of the following control levels for the pre-planned images:
699: Maximum control by pre-planned images.
799: High control by pre-planned images.
899: Moderate control by pre-planned images.
999: Minimal control by pre-planned images.

Here're some examples:
{In-Context Learning Example}

User Input: {Prompt}

Figure 7: Prompt template used to determine how much noise to inject during inversion.

over text-to-video generation in a zero-shot setting, allowing users to more precisely direct the motion of objects within the video.

However, VIDEO-MSG has several limitations. It relies on a strong instruction-following LLM to generate planning sequences from user input. While it can be implemented using publicly available models such as Qwen2-VL, stronger models like GPT-4o yield better performance. Besides, although VIDEO-MSG improves text-to-video generation in terms of object motion, spatial relationships, and numeracy, it does not guarantee strict adherence to every detail in the text prompt. Therefore, careful evaluation is necessary before applying VIDEO-MSG in domain-specific scenarios.

## D LICENSES

We provide the licenses of the existing assets we use in this paper in table 5.

Table 5: A list of the licenses of the existing assets used in this paper.

| Asset | License |
|---|---|
| PyTorch (Ansel et al., 2024) | BSD-style |
| Huggingface Transformers (Wolf et al., 2020) | Apache License 2.0 |
| Torchvision (maintainers & contributors, 2016) | BSD 3-Clause "New" or "Revised" License |
| Diffusers (von Platen et al., 2022) | Apache License 2.0 |
| VideoCrafter2 (Chen et al., 2024) | Apache License 2.0 |
| CogVideoX (Yang et al., 2024b) | Apache License 2.0 |
| GPT4-o (OpenAI et al., 2024) | OpenAI Terms of Use |
| Qwen2-VL (Bai et al., 2023) | Tongyi Qianwen LICENSE AGREEMENT |

