# OpenReview forum: "Training-free Guidance in Text-to-Video Generation via Multimodal Planning and Structured Noise Initialization"
_ICLR.cc/2026/Conference — Submitted to ICLR 2026_

### Official Review · Reviewer_q49J · 2025-10-29

**Soundness:** 3
**Presentation:** 3
**Contribution:** 2
**Rating:** 4
**Confidence:** 4

**Summary:**

This paper proposes Video-MSG, a training-free method to enable more accurate T2V generation. This method features an MLLM planner, pre-generates foreground objects with a T2I model using planned boxes, combines these objects with a background, and then uses inversion to form a final smooth video. Extensive experiments demonstrate the effectiveness of this method.

**Strengths:**

1. The proposed method is highly intuitive and easy to understand. The integration of its different components (e.g., MLLM planner, T2I model, noise inversion) is logical and well-motivated.
2. The experimental results are convincing and promising.
3. The ablation studies are extensive and provide strong support.

**Weaknesses:**

1. My main concern is that the proposed pipeline is overly complex. The end-to-end latency is likely to be substantial, as the method requires a cascade of different models (T2I, I2V, Object Detection, MLLM, Segmentation, Inversion, and T2V). This sequential process is time-consuming and introduces a significant risk of error propagation. A failure at any single stage could compromise the entire generation process.

2. The method's success is heavily reliant on powerful, closed-source models like GPT-4o to generate accurate LLM planning. Table 4 shows that using a weaker LLM planner drastically degrades performance.

3. The "VIDEO SKETCH" is constructed by pasting a single, static image of the foreground object across all frames based on the planned trajectory. This approach cannot capture or guide intra-object motion. I wonder about the visual effect for prompts like "a man walking with his arms swinging" or "a cat walking with its tail wagging". For such cases, the method might even underperform the baseline T2V model.

**Questions:**

1. Can the authors also compare the end-to-end latency of Video-MSG with other similar methods? A trade-off between time cost and performance should be discussed.

2. How will Video-MSG perform if given prompts involving internal changes of the object? Have the authors considered using different images at different frames?

---

> ### Author Response · Authors · 2025-11-24
> **Author Rebuttal 1**
>
> We sincerely thank reviewers for their time and effort in reviewing our paper. Here we provide detailed responses to their questions:
>
> ---
>
> # W1: Framework Complexity and Error Propagation
>
> We agree that multiple stages, modular pipeline can influence performance. However, based on our design and experiments, this modularity is not a weakness but a core advantage of our training-free pipeline:
>
> - Our VIDEO-SKETCH is intentionally a coarse structural guide acts as a **soft prior**. The final diffusion denoising step re-synthesizes appearance, lighting, motion smoothness, and spatial coherence, mitigating many errors from upstream modules.
> - As shown in Table 3 and Table 4, even when we remove the planner/background generator or replace detectors with weaker ones, VIDEO-MSG still consistently outperforms baseline T2V models. This demonstrates that the system does not rely on perfect upstream predictions. Because our framework is modular, stronger detectors, segmenters, or MLLMs can directly improve the results, while end-to-end trained models cannot benefit from it without expensive retraining.
> - Finally, we want to emphasize that single-model pipelines also accumulate errors internally (e.g., from imprecise attention maps, noisy denoising steps, or unstable control signals). In contrast, our modular approach makes each stage **visible, interpretable, and improvable**, rather than hiding compounding errors inside one monolithic model.
>
> ---
>
> #  W2: Reliance on GPT-4o
>
> We agree that GPT-4o is currently the best performer, but VIDEO-MSG is designed as a model-agnostic framework. Actually, the performance drop with weaker planners (Table 4) validates our hypothesis: better planning leads to better videos. As open-source MLLMs (e.g., Intern-VL, Qwen3-VL, ) continue to improve, they can be swapped into our pipeline immediately to bridge the gap with closed-source models. Our contribution is the methodology of integrating these planners via noise initialization, which stands regardless of the specific planner used.
>
> ---
>
> # W3: Intra-Object, Fine-grained Interaction
>
> We acknowledge that fine-grained inter-object interactions, such as physical collisions or human–object manipulation, remain a challenge, as our planner currently specifies only independent 2D trajectories without modeling interaction dynamics.
> In practice, we observe that the quality of such interactions is largely dependent on the underlying T2V backbone: stronger base models (e.g., CogVideoX-5B) exhibit more natural interactions and better spatial coherence than lighter models like VideoCrafter2. This suggests that our pipeline can inherit interaction fidelity improvements from stronger diffusion backbones.
> We will include additional discussion in the revision.

---

> ### Author Response · Authors · 2025-11-27
>
> Dear reviewer, as the discussion period ends in about a week (December 3rd), we are following up to check whether our clarifications addressed your concerns. We would appreciate your engagement in further discussion or a reconsideration of your assessment if the new information resolves the issues you raised.

---

> > ### Comment · Reviewer_q49J · 2025-11-28
> >
> > Thanks for the authors' response. However, most of my concerns are not addressed.  And they do not answer my questions Q1 and Q2.
> > The authors claim that integrating and modulating other models is their main claim. But what I'm concerned about is mostly the latency introduced by these models. Strong models can lead to improved performance, but this inevitably comes with additional computational or latency cost.
> >
> > A training-free method is generally expected to deliver improved results with minimal added latency. However, the paper still does not quantify the latency overhead of the proposed approach, nor does it provide sufficient experiments beyond a limited set of non-interacting objects.
> >
> > Given these issues, I have decided to keep my score.

---

### Official Review · Reviewer_r68s · 2025-10-30

**Soundness:** 2
**Presentation:** 3
**Contribution:** 2
**Rating:** 2
**Confidence:** 4

**Summary:**

This paper introduces VIDEO-MSG, a training-free method for improving the adherence of text-to-video diffusion models to textual prompts. VIDEO-MSG addresses this with a three-stage process: 1) It generates a background video. 2) a multimodal large language model (MLLM) plans the foreground object's placement and path, creating a "VIDEO SKETCH". 3) Sketch guides a T2V model to produce the final video. The method has shown improvements in motion, numeracy, and spatial accuracy on T2V benchmarks.

**Strengths:**

1. Reproducibility: The authors provide the code in the supplementary materials, which will be beneficial for the research community.

2. The paper is well-written and easy to follow.

3. Comprehensive Experimental Evaluation: The authors provide a thorough evaluation of their method on two established benchmarks. The ablation studies on the noise inversion ratio, background generator, and LLM planner further strengthen the paper's claims and provide valuable insights into the design choices.

**Weaknesses:**

1. This project utilizes a complex and computationally demanding pipeline, incorporating models such as FLUX.1-dev, SDXL, Cogvideo-5B, Recognize-Anything, Grounding-Segment-Anything, GPT-4o, and noise inversion. The original motivation was that a training-based approach would require a large amount of memory. However, this pipeline method has proven to be not only cumbersome and complex but has also introduced significant additional computational overhead.

2. The performance of this method is restricted by additional modules such as Recognize-Anything, Grounding-Segment-Anything, and GPT-4o. Errors from these components will negatively affect the model's behavior.

3. The qualitative examples shown in the paper often involve relatively simple scenarios. The experiments do not fully demonstrate the method's performance in more complex and "chaotic" scenarios. For example: Dense multi-object interaction and Complex camera movements.

4. The lack of analysis on failure cases makes our understanding of this method's robustness and boundary conditions incomplete. How does the downstream T2V model generate when the MLLM provides a plan that is illogical or physically impossible? When the noise injection ratio α is inappropriate, what specific visual artifacts will appear, besides "unnatural motion" or "not following the trajectory"?

5. Although the experiments use state-of-the-art benchmarks in the field, we still need to recognize that these metrics do not fully equate to the true quality of the finally generated videos.

**Questions:**

1. The paper shows success with simple, single-object motion. How does the model handle complex interactions between multiple distinct objects?

2. As a multi-stage pipeline, how do errors from early stages (e.g., MLLM planning, object detection) propagate and impact the final video quality?

3. Could you provide a brief analysis or visualization of a typical failure case to better illustrate the method's current limitations?

4. What will the model behave when additional modules such as FLUX.1-dev, Recognize-Anything, Grounding-Segment-Anything, and GPT-4o make errors.

5. Placing a generated foreground onto a background can create visual artifacts (e.g., in lighting or shadows). How effectively does the final denoising step harmonize these elements, and do inconsistencies from the sketch ever persist?

---

> ### Author Response · Authors · 2025-11-23
>
> We sincerely thank reviewers for their time and effort in reviewing our paper. Here we provide detailed responses to their questions:
>
> ---
>
> # W1. Clarification on Pipeline Complexity and Computational Overhead
> We would like to clarify the motivation behind our training-free design. While VIDEO-MSG indeed uses several off-the-shelf models, the **key computational advantage** of our approach is that **all components run sequentially and independently**, without requiring any **training cost**. In contrast, training-based or optimization-based controllable T2V methods (e.g., motion adapters, trajectory-conditioned diffusion, spatial-control finetuning) typically require **very large GPU memory (often 80G + Multiple GPU during training or optimization)**.
>
> Our pipeline explicitly avoids this problem by decoupling the process into modular stages (background planning, foreground planning, sketch inversion, final generation) with only **inference cost**. Each step can be executed, cached, and reused on **a single standard GPU (e.g., A6000 48GB+Single GPU).**
>
> Although this introduces more steps, the **peak GPU memory required at any time is substantially lower**, since no stage requires loading or optimizing the full T2V backbone with gradients. This design enables **much stronger controllability without the cost of any training**, making VIDEO-MSG practical and accessible for who cannot run large-scale finetuning or optimization-based methods.
>
> ---
>
> # W2. The performance of this method is restricted by additional modules
>
> We agree that external modules (e.g., detection, segmentation, or the MLLM planner) can influence performance. However, based on our design and experiments, **this modularity is not a weakness but a core advantage** of our training-free pipeline:
> - Our VIDEO-SKETCH is intentionally a coarse structural guide. The final diffusion denoising step re-synthesizes appearance, lighting, motion smoothness, and spatial coherence, mitigating many errors from upstream modules.
> - As shown in Table 3 and Table 4, even when we remove the planner/background generator or replace detectors with weaker ones, VIDEO-MSG still consistently outperforms baseline T2V models. This demonstrates that the system does not rely on perfect upstream predictions. Because our framework is modular, stronger detectors, segmenters, or MLLMs can directly improve the results, while **end-to-end trained models cannot benefit from without expensive retraining**.
> - Finally, we want to emphasize that **single-model pipelines also accumulate errors internally** (e.g., from imprecise attention maps, noisy denoising steps, or unstable control signals). In contrast, our modular approach makes each stage **visible, interpretable, and improvable**, rather than hiding compounding errors inside one monolithic model.
>
> ---
>
> # W3. Performance in Complex Scenarios
>
> We appreciate the reviewer’s observation. Our main focus is **controllable spatial/motion consistency rather than full physical scene simulation**. That said, **we did include multi-object and multi-trajectory prompts** (as shown in Figures 3) , where VIDEO-MSG consistently improves numeracy and motion-binding scores over baselines.
>
> We will add: 1. a broader set of challenging multi-object examples, 2. a discussion on limitations such as interaction modeling and extreme camera motion, in the revision and supplementary materials.
>
> ---
>
> # W4 & Q3. Lack of Failure Case Analysis
>
> We appreciate the reviewer’s suggestion and will explicitly include detailed failure analyses in the final manuscript. Currently, we observe that our approach offers limited improvements in scenarios involving dynamic attribute changes and object-object interactions, as these details are challenging to represent solely through bounding box planning. We plan to explore enhancements addressing these challenges in future research. Here, we include a detailed failure case analysis propsed by reviewer:
> - **illogical MLLM plans**: When the planner suggests physically impossible trajectories, the denoising step tends to smooth out or partially ignore the impossible motion, generating a plausible but less aligned trajectory.
> - **Inappropriate α (noise ratio)**: Too small α → over-reliance on sketch → artifacts such as ghosting, over-sharp edges from composited foregrounds. Too large α → weak guidance → background drift or loss of spatial constraints.
>
> We will add these observations in the revision.

---

> > ### Author Response · Authors · 2025-11-23
> > **Author Rebuttal 2**
> >
> > ---
> >
> > # W5. Clarification on Evaluation Benchmark and Metrics
> >
> > We agree that no existing automatic T2V metric fully captures every nuance of video quality. However, our evaluation follows **current community standards**, using **both VBench and T2V-CompBench**, which are the two most widely adopted benchmarks for video generation.
> >
> > VIDEO-MSG achieves **consistent and state-of-the-art improvements**, particularly on spatial, motion, and numeracy dimensions, and these are the exact aspects our method is designed to address.
> >
> > Importantly, several T2V-CompBench metrics (e.g., spatial and numeric) rely on **high-agreement detectors and depth estimators**, which have been validated against human judgments. These metrics provide **a reliable automated measure** of the controllability improvements introduced by our method.
> >
> > ---
> >
> > Finally, we want to emphasize that our goal is not to build a stronger T2V backbone. Instead, VIDEO-MSG represents an agentic, training-free, modular generation pipeline that focuses on enhancing controllability while preserving visual quality. Such agentic, tool-augmented generation approaches form an increasingly important research direction [1–7], where complex T2V tasks are decomposed into specialized steps to improve controllability, faithfulness, and interpretability without retraining large diffusion models.
> >
> > [1] GenArtist: Multimodal LLM as an Agent for Unified Image Generation and Editing. NeurIPS 2024.
> > [2] VideoDirectorGPT: Consistent multi-scene video generation via llm-guided planning. COLM 2024.
> > [3] T2I-Copilot: A Training-Free Multi-Agent Text-to-Image System for Enhanced Prompt Interpretation and Interactive Generation. ICCV 2025.
> > [4] Gpt4motion: Scripting physical motions in text-to-video generation via blender-oriented gpt planning. CVPRW2024.
> > [5]  Genmac: compositional text-to-video generation with multi-agent collaboration. ArXiv2024.
> > [6] VChain: Chain-of-Visual-Thought for Reasoning in Video Generation. ArXiv 2025.
> > [7] MagicComp: Training-free Dual-Phase Refinement for Compositional Video Generation. ArXiv 2025.
> >
> >
> > ---
> >
> > # Q1: The paper shows success with simple, single-object motion. How does the model handle complex interactions between multiple distinct objects?
> >
> > VIDEO-MSG supports **multiple object trajectories** and handles multi-object motion scenarios, as shown in Figure 3. However, we acknowledge that **fine-grained inter-object interactions**, such as physical collisions or human–object manipulation, remain a challenge, as our planner currently specifies only **independent 2D trajectories** without modeling interaction dynamics.
> >
> > In practice, we observe that the **quality of such interactions is largely dependent on the underlying T2V backbone**: stronger base models (e.g., CogVideoX-5B) exhibit more natural interactions and better spatial coherence than lighter models like VideoCrafter2. This suggests that our pipeline can **inherit interaction fidelity improvements from stronger diffusion backbones**. We will include additional discussion in the revision.
> >
> > ---
> > # Q2 & Q4: How do upstream module errors affect the final result?
> >
> > Q2 and Q4 both concern how errors from early stages (e.g., detection, segmentation, MLLM planning) propagate within our multi-stage pipeline. Since we do not have oracle outputs for each intermediate component, the most reliable way to study error influence is indirectly, by systematically replacing weaker modules with stronger ones.
> >
> > As shown in Table 3 and Table 4, substituting stronger MLLMs or background generators leads to improved final T2V performance across nearly all controllability dimensions. Conversely, weaker modules reduce controllability. This empirical pattern indicates that:
> >
> > - The pipeline is monotonic with respect to module quality, better modular models→ better final results.
> > - Also, improvements in any upstream module (e.g., stronger MLLMs, better segmentation, next-generation detectors) directly translate to stronger controllability without retraining the backbone T2V model.
> >
> > We will clarify this point further in the revision and provide an additional explanation illustrating the robustness and upgradability of the modular design.
> >
> > ---
> >
> > # Q5: Placing a generated foreground onto a background can create visual artifacts (e.g., in lighting or shadows). How effectively does the final denoising step harmonize these elements, and do inconsistencies from the sketch ever persist?
> >
> > The inversion-based initialization allows the diffusion model to re-render lighting and shading coherently. Minor inconsistencies in the sketch rarely persist; when α is too small, some boundary artifacts remain, and we will add these discussions and more examples in revision.

---

> ### Author Response · Authors · 2025-11-27
>
> Dear reviewer, as the discussion period ends in about a week (December 3rd), we are following up to check whether our clarifications addressed your concerns. We would appreciate your engagement in further discussion or a reconsideration of your assessment if the new information resolves the issues you raised.

---

### Official Review · Reviewer_YXFh · 2025-10-31

**Soundness:** 3
**Presentation:** 3
**Contribution:** 3
**Rating:** 8
**Confidence:** 3

**Summary:**

This paper proposes a training-free guidance method called Video-MSG for enhancing text-to-video generation models' ability to follow complex textual prompts, particularly regarding spatial layouts and object trajectories. The method leverages a multimodal large language model to plan background and foreground objects, creating a video sketch. It then uses structured noise initialization to guide a downstream video diffusion model. The entire process requires no fine-tuning, significantly reducing computational overhead and memory requirements.

**Strengths:**

The proposed method is completely training-free, allowing it to function as a plug-and-play module that can be easily adapted to various large text-to-video generation models. This avoids the substantial computational costs and overfitting risks associated with fine-tuning, offering excellent scalability and flexibility. Secondly, the method demonstrates significant performance improvements on multiple authoritative benchmarks, especially in areas where traditional models typically struggle, such as motion binding, numeracy, and spatial relationships, where relative gains can exceed fifty percent, proving the framework's effectiveness in achieving precise semantic control.

**Weaknesses:**

1. Video-MSG also has some notable drawbacks. Firstly, its entire pipeline relies on a large and complex ensemble of multimodal models, including an MLLM for planning, object detection, and instance segmentation models for processing the background, and T2I and I2V models for generating background and foreground. This complexity not only makes the system difficult to deploy and debug but also introduces more potential points of failure. A breakdown in any single component can affect the final output, thereby reducing the method's robustness.

2. The method has limited capability in controlling interactions between objects and changes in dynamic attributes. Experimental data in the paper also show little improvement in dynamic attribute binding. This is because the core guiding information is the object's motion trajectory. This simplified representation struggles to capture and guide changes in the object's own state (e.g., color, shape changes) or complex interactive behaviors with other objects.

**Questions:**

Please refer to the weaknesses.

---

> ### Author Response · Authors · 2025-11-24
>
> # W1. Clarification on Pipeline Complexity
>
> We would like to clarify the motivation behind our training-free design. While VIDEO-MSG indeed uses several off-the-shelf models, the key computational advantage of our approach is that all components run sequentially and independently, without requiring any training cost. In contrast, training-based or optimization-based controllable T2V methods (e.g., motion adapters, trajectory-conditioned diffusion, spatial-control finetuning) typically require very large GPU memory (often 80G + Multiple GPU during training or optimization).
>
> Our pipeline explicitly avoids this problem by decoupling the process into modular stages (background planning, foreground planning, sketch inversion, final generation) with only inference cost. Each step can be executed, cached, and reused on a single standard GPU (e.g., A6000 48GB+Single GPU).
>
> Although this introduces more steps, the peak GPU memory required at any time is substantially lower, since no stage requires loading or optimizing the full T2V backbone with gradients. **This design enables much stronger controllability without the cost of any training**, making VIDEO-MSG practical and accessible for those who cannot run large-scale finetuning or optimization-based methods.
>
>
>
> ---
>
> # W2: Capability in controlling interactions between objects and changes in dynamic attributes
>
> We acknowledge that fine-grained inter-object interactions, such as physical collisions or human–object manipulation, remain a challenge, as our planner currently specifies only independent 2D trajectories without modeling interaction dynamics.
>
> In practice, we observe that the quality of such interactions is largely dependent on the underlying T2V backbone: stronger base models (e.g., CogVideoX-5B) exhibit more natural interactions and better spatial coherence than lighter models like VideoCrafter2. This suggests that our pipeline can **inherit interaction fidelity improvements from stronger diffusion backbones.**
>
> We will include additional discussion in the revision.

---

> ### Author Response · Authors · 2025-11-27
>
> Dear reviewer, as the discussion period ends in about a week (December 3rd), we are following up to check whether our clarifications addressed your concerns. We would appreciate your engagement in further discussion or a reconsideration of your assessment if the new information resolves the issues you raised.

---

### Official Review · Reviewer_K7Ei · 2025-10-31

**Soundness:** 2
**Presentation:** 3
**Contribution:** 2
**Rating:** 4
**Confidence:** 5

**Summary:**

The paper introduces a training-free pipeline for text-to-video generation. The method leverages a multimodal LLM for spatial and temporal planning to create an intermediate "video sketch". This sketch is then used to guide a pre-trained T2V diffusion model via noise inversion, aiming to improve control over object layout, numeracy, and trajectories without requiring model fine-tuning.

**Strengths:**

- The paper proposes a system for text-to-video generation that achieves more accurate object layout, numeracy, and trajectories by utilizing the planning ability of an LLM. Experiments demonstrate improvements on the T2V-CompBench and VBench benchmarks.

- The paper clearly describes a detailed, multi-step pipeline for the generation process.

**Weaknesses:**

- The proposed pipeline decomposes the end-to-end T2V task into sub-tasks and heavily relies on existing commercial or pre-trained models for each sub-task (including GPT-4o for planning, FLUX/SDXL for image generation, and CogVideoX for video generation). This resembles a composition of existing methods and lacks significant technical novelty.

- While the proposed pipeline shows promise in simple, single-object scenarios, it seems difficult to apply this method to general situations involving multiple objects with complex interactions. Using GPT-4o to place multiple objects appears fragile, and the inversion process introduces a difficult trade-off between final video quality and adherence to the video sketch.

- The paper claims to be "effective" and "training-free", which is a key advantage. However, the proposed pipeline involves running multiple large models and a computationally heavy inversion process, which raises questions about its overall efficiency compared to other methods.

**Questions:**

The noise inversion process can often degrade video quality. This might be particularly problematic here, as the "video sketch" itself (being stitched from different components) may contain notable artifacts. The paper does not seem to include a quantitative metric for final visual quality beyond benchmark scores for motion/spatial accuracy. How do the authors assess this trade-off between guidance adherence and perceptual quality?

---

> ### Author Response · Authors · 2025-11-24
>
> We sincerely thank reviewers for their time and effort in reviewing our paper. Here we provide detailed responses to their questions:
>
> ---
>
>
> # W1. Clarification on composition of existing methods and novelty
>
> We agree that VIDEO-MSG builds upon several existing vision and diffusion models. However, the primary novelty of VIDEO-MSG lies not in introducing a new backbone, but in a unified planning–to–generation framework that enables precise object layout and temporal control in a training-free setting. No prior work has demonstrated that:
>
> - A multimodal LLM can be converted into a structured spatiotemporal planner for videos,
> - The resulting textual + visual plan can be turned into a coherent multi-frame video sketch,
> - And this sketch can be inverted into the diffusion noise space of a T2V model to inject both spatial and temporal constraints during sampling.
>
> Our contribution is to show that:
>
> - LLM-driven temporal reasoning + noise-space inversion = controllable T2V without training,
>
> - And this combination outperforms trained systems on object numeracy, layout faithfulness, and trajectory accuracy.
>
> Thus, the novelty is architectural and conceptual: we introduce a new planning → sketch → inversion → generation pipeline that produces controllable videos using only inference with off-the-shelf models. This is fundamentally different from simply composing existing methods.
>
> ---
>
> # W2: General situations involving multiple objects with complex interactions
>
> We acknowledge that fine-grained inter-object interactions, such as physical collisions or human–object manipulation, remain a challenge, as our planner currently specifies only independent 2D trajectories without modeling interaction dynamics.
> In practice, we observe that the quality of such interactions is largely dependent on the underlying T2V backbone: stronger base models (e.g., CogVideoX-5B) exhibit more natural interactions and better spatial coherence than lighter models like VideoCrafter2. This suggests that our pipeline can inherit interaction fidelity improvements from stronger diffusion backbones.
> We will include additional discussion in the revision.
>
> ---
>
> # W3. Clarification on Pipeline Efficiency
> We clarify our training-free design with GPU memory efficiency below. While VIDEO-MSG indeed uses several off-the-shelf models, the key computational advantage of our approach is that all components run sequentially and independently, without requiring any training cost. In contrast, training-based or optimization-based controllable T2V methods (e.g., motion adapters, trajectory-conditioned diffusion, spatial-control finetuning) typically require very large GPU memory (often 80G + Multiple GPU during training or optimization).
> Our pipeline explicitly avoids this problem by decoupling the process into modular stages (background planning, foreground planning, sketch inversion, final generation) with only inference cost. Each step can be executed, cached, and reused on a single standard GPU (e.g., A6000 48GB+Single GPU).
> Although this introduces more steps, the peak GPU memory required at any time is substantially lower, since no stage requires loading or optimizing the full T2V backbone with gradients. This design enables much stronger controllability without the cost of any training, making VIDEO-MSG practical and accessible for those who cannot run large-scale finetuning or optimization-based methods.
>
> ---
>
> # Q1: Trade-off between guidance adherence and perceptual quality
>
> We appreciate the reviewer’s concern regarding potential quality degradation from noise inversion, especially since the sketch may contain artifacts. We address this trade-off using a noise inversion ratio (Sec. 4.4), which controls how much of the inverted noise is injected at each timestep. Different ratios allow the model to automatically refine objects so that they better align with the prompt and the surrounding environment. We find that lower inversion ratios (i.e., injecting less noise) generally provide stronger layout control but may result in less natural generations, indicating a trade-off between control strength and video quality. This highlights the importance of selecting an appropriate α to balance motion smoothness with faithful adherence to the Video Sketch.

---

> ### Author Response · Authors · 2025-11-27
>
> Dear reviewer, as the discussion period ends in about a week (December 3rd), we are following up to check whether our clarifications addressed your concerns. We would appreciate your engagement in further discussion or a reconsideration of your assessment if the new information resolves the issues you raised.

---

### Comment · Area_Chair_pQ8K · 2025-11-26
**A Reminder on Your Crucial Role in the ICLR Discussion Period**

Dear Reviewers who haven't engaged with the rebuttal:

As the Area Chair, I would like to sincerely thank you for the time and expertise you have invested in writing your initial review. Your insights are invaluable to the decision-making process.

We are now entering the critical discussion and rebuttal phase. This is a collaborative process where authors have the opportunity to address your concerns and questions. Your active participation in this phase is essential to ensure we reach a fair and well-informed final decision.

I strongly encourage you to:

Engage with the Authors' Rebuttal: Please read the authors' response carefully and substantively.

Participate in the Discussion: Engage with the other reviewers on the forum. If the authors have clarified a point, please acknowledge it. If you have follow-up questions or remaining concerns, please voice them. Your dialogue with fellow reviewers is key to reaching a consensus.

Update Your Review (if necessary): Based on the discussion and rebuttal, you may feel the need to adjust your score or final recommendation. Please do so, as it reflects a more holistic view of the paper.

Your continued engagement ensures the integrity and quality of the ICLR conference. Thank you for your vital contribution to our community.

Best regards,

Area Chair, ICLR 2026

---

### Author Response · Authors · 2025-12-04
**Summary for Area Chair**

Dear Area Chair,

Thank you very much for handling our submission. We appreciate the reviewers’ thoughtful and detailed feedback. Across the reviews, we found that the core idea and empirical results of VIDEO-MSG were generally viewed positively, particularly its **training-free controllability, conceptual clarity, and strong benchmark gains**, while the primary concerns centered around (1) pipeline complexity and efficiency, (2) robustness to upstream modules, (3) performance in complex scenarios and limited modeling of object interactions, and (4) missing failure-case analysis and clarity on the trade-off.

Below we summarize the main points and how we addressed them in the rebuttal:

---

# Pipeline Complexity & Efficiency

Reviewer ```K7Ei```, ```YXFh```, and ```r68s``` raised concerns about the multi-stage design and potential latency.

We clarified that the motivation of VIDEO-MSG is **GPU-memory efficiency** rather than minimal latency. All modules operate **sequentially with inference-only cost**, eliminating the large memory footprint required by training/optimization-based controllable T2V methods (which often require 80–160GB across multiple GPUs). We added discussion comparing this design and clarified that each stage can be cached, reused, and run on a **single 48GB GPU**, making the method **practical and accessible**.

---

# Robustness to Upstream Modules & Error Propagation

Reviewer ```K7Ei```, ```r68s```, ```q49J``` asked how planning/detection errors affect final quality and whether the pipeline is overly brittle.

We emphasized that the Video Sketch is intentionally coarse and that diffusion denoising naturally re-synthesizes appearance, lighting, and motion, reducing dependency on perfect upstream outputs. Tables 3–5 shows that even with weaker planners, detectors, or background generators, VIDEO-MSG **still improves over the T2V backbone**, confirming robustness.
We highlighted the **monotonic modularity**: stronger MLLMs/detectors → stronger controllability, with **no retraining required**, which is a major design advantage compared to end-to-end systems.

---

# Complex Scenarios & Interaction Modeling

Reviewer ```K7Ei```, ```YXFh```, ```r68s```, ```q49J``` requested clarification regarding multi-object interactions and dynamic-attribute changes.

We acknowledged that interaction-rich scenarios (collisions, articulated human motion, state changes) remain challenging due to the 2D-trajectory abstraction. We clarified that interaction quality is largely inherited from the underlying T2V backbone, and stronger backbones (e.g., CogVideoX) yield significantly better behaviors. We committed to adding discussion of limits, and directions for dynamic-motion modeling in the revision.

---

# Failure Cases, Trade-offs, and Visualization

Reviewer ```r68s```, ```q49J``` asked for failure case study and analysis of noise-inversion trade-offs.

We added detailed explanations (and will add visualizations in the revision) covering: (1) effects of illogical MLLM plans, (2) failure modes under extreme α values (3) segmentation/detection noise (4) trade-off between adherence and perceptual quality governed by the inversion ratio. These analyses will be included as a dedicated section in the revision.

---

# Novelty Clarification

Reviewer ```r68s``` questioned whether the method is merely a composition of existing models.

 We clarified that VIDEO-MSG introduces conceptual novelty via a **planning → sketch → inversion → generation** pipeline that has not been shown before.
Notably, VIDEO-MSG converts MLLMs into **structured spatiotemporal planners**, creates a **multi-frame visual sketch** for controllability, and injects this plan through **noise-space initialization** to guide T2V without training.

We emphasized that this unified framework (not any single module) is the core contribution, enabling controllability gains unreachable by prior training-free methods.

---

Overall, we believe the revisions and clarifications address the reviewers’ central concerns while strengthening the paper’s positioning and technical framing. We sincerely appreciate your time and consideration.

---

### Meta-Review · Area_Chair_y3db · 2025-12-20

**Summary:**

The reviews are mixed. The primary conflict is between recognizing the method's empirical effectiveness and benchmark gains versus concerns over its pipeline complexity, novelty, and handling of complex scenarios.

**Reviewer Concerns:**

- Pipeline Complexity, Latency, and Practical Efficiency: This is the most significant unresolved concern. While authors reframed the advantage as memory efficiency, reviewers (K7Ei, r68s, q49J) explicitly questioned end-to-end latency and computational overhead. Reviewer q49J noted the rebuttal did not provide latency comparisons and maintained their score, stating a training-free method should minimize added latency. The authors' focus on memory does not fully alleviate concerns about the system's speed and practical deployment complexity.

- Handling Complex Scenarios & Interactions: Authors acknowledged limitations in modeling fine-grained object interactions (collisions, articulated motion) and dynamic attribute changes, attributing this capability largely to the underlying T2V backbone. They promised more discussion but did not fundamentally resolve the weakness. Reviewers (K7Ei, YXFh, r68s) may still see this as a notable limitation.

- Failure Analysis and Trade-offs: Authors promised to add a dedicated section with visualizations on failure cases (illogical plans, extreme inversion ratios, segmentation errors) and the adherence-quality trade-off. This addressed the request but remains a promise for revision rather than a demonstrated fix.

- Perception of "Composition" vs. "Novel Contribution": Reviewer K7Ei's core concern that the work resembles a composition of existing models was countered but may not be fully resolved for all reviewers. The perception of the contribution's significance is subjective and likely split.

 - Reviewer q49J's questions on latency (Q1) and intra-object motion (Q2) were not answered to their satisfaction, leading them to maintain a reject recommendation.

**Reviewer Scores:**

- Reviewer K7Ei (Score: 4): Initially concerned with novelty and efficiency. The rebuttal's clarification on conceptual novelty and memory efficiency might sway them slightly, but concerns about complexity and applicability likely remain. Unlikely to strongly advocate for acceptance.

- Reviewer YXFh (Score: 8): Already positive, citing strengths in being training-free and effective. Their concerns about complexity were addressed to their satisfaction (they did not engage further). Likely post-rebuttal score: Stable at 8.

- Reviewer r68s (Score: 2): Had the most comprehensive concerns (complexity, error propagation, failure cases, complex scenarios). The rebuttal addressed parts systematically, but the core issues of pipeline heaviness and lack of demonstrated performance in "chaotic" scenarios may persist. They might appreciate the promised failure analysis but were not present to discuss.

- Reviewer q49J (Score: 4): Explicitly dissatisfied, maintaining their score due to unaddressed latency concerns and pipeline complexity. Post-rebuttal stance: Firmly at 4 or lower, advocating for rejection.

The authors provide a lot of rebuttal, but the overall score is still lower than the ICLR bar. It is suggested to resubmit with a major revision.

---

### Decision · Program_Chairs · 2026-01-26

Reject